# Emergence of ferroelectricity in Sn-based perovskite semiconductor films by iminazole molecular reconfiguration

Yu Liu[1,2], Shuzhang Yang[2], Lina Hua[1], Xiaomin Yang[1], Enlong Li[1], Jincheng Wen[1], Yanqiu Wu[1], Liping Zhu[3], Yingguo Yang [4], Yan Zhao [1], Zhenghua An [3], Junhao Chu[1,2] & Wenwu Li [1,2] ✉

Ferroelectric semiconductors have the advantages of switchable polarization ferroelectric field regulation and semiconductor transport characteristics, which are highly promising in ferroelectric transistors and nonvolatile memory. However, it is difficult to prepare a Sn-based perovskite film with both robust ferroelectric and semiconductor properties. Here, by doping with 2-methylbenzimidazole, Sn-based perovskite [93.3 mol% $(FA_{0.86}Cs_{0.14})SnI_3$ and 6.7 mol% $PEA_2SnI_4$] semiconductor films are transformed into ferroelectric semiconductor films, owing to molecular reconfiguration. The reconfigured ferroelectric semiconductors exhibit a high remanent polarization ($P_r$) of 23.2 $\mu C/cm^2$. The emergence of ferroelectricity can be ascribed to the hydrogen bond enhancement after imidazole molecular doping, and then the spatial symmetry breaks causing the positive and negative charge centers to become non-coincident. Remarkably, the transistors based on perovskite ferroelectric semiconductors have a low subthreshold swing of 67 mv/dec, which further substantiates the superiority of introducing ferroelectricity. This work has developed a method to realize Sn-based ferroelectric semiconductor films for electronic device applications.

Ferroelectric materials have spontaneous polarization and electric-field switchable macroscopic polarization properties, owing to the noncoincidence between positive and negative charge centers[1,2]. Primarily, ferroelectric materials can be classified into insulators ($Hf_{0.5}Zr_{0.5}O_2$[3], $BaTiO_3$[4], P(VDF-TeFe)[5], $LiNbO_3$[6], etc.) and semiconductors (α-$In_2Se_3$[7], β-$CuGaO_2$[8], SnS[9], etc.) based on their bandgap energy. When the bandgap >3 eV, the ferroelectric materials are generally considered insulators, which can act as dielectric materials[10]. On the other hand, owing to the additional nonvolatile ferroelectric field, the carrier transport ability is promoted, making ferroelectric semiconductors useful for low-power electronics and nonvolatile memory[11,12]. However, the realization of a thin film exhibiting both robust ferroelectricity and exceptional semiconductor properties remains a considerable challenge.

In recent years, Sn-based perovskite semiconductors have received much attention owing to their p-type characteristics and lower effective mass of carriers[13]. The strong anti-bonding coupling between Sn 5s and I 5p orbitals lowers the formation energy of Sn vacancies, leading to high hole carrier density[14]. Remarkably, ref. 15 reported a p-type Sn-based perovskite transistor with a high mobility of 72 $cm^2 v^{-1} s^{-1}$. However, Sn-based perovskite semiconductors with strong ferroelectricity are rarely reported. Due to the high carrier concentration, the internal electric field cannot be effectively screened, leading to weakened or eliminated ferroelectric

[1]State Key Laboratory of Photovoltaic Science and Technology, Department of Materials Science, Institute of Optoelectronics, Fudan University, Shanghai 200433, China. [2]Shanghai Frontiers Science Research Base of Intelligent Optoelectronics and Perception, Fudan University, Shanghai 200433, China. [3]State Key Laboratory of Surface Physics, Institute for Nanoelectronic Devices and Quantum Computing, Department of Physics, Fudan University, Shanghai 200433, China. [4]School of Microelectronics, Fudan University, Shanghai 200433, China. ✉e-mail: liwenwu@fudan.edu.cn

polarization. High carrier concentration can screen the ferroelectric polarization field, thereby hindering the observation of ferroelectricity. The conventional remnant P − E hysteresis loop of $\alpha$-In$_2$Se$_3$ is difficult to demonstrate by the Sawyer–Tower method due to charge screening[12]. Thicker SnS films exhibit a markedly diminished ferroelectric response since the screening effects are induced by the higher concentration of charge carriers[16]. The reason behind this phenomenon is studied using (Sr, Ca, or Ba)TiO$_3$ with doping ions, who discovered (Sr, Ca, or Ba)TiO$_3$ introduces carriers through Nb$^{5+}$ ion doping, with the doping concentration ranging from -0.05 to 0.5%. Within this range, the resistance decreases while ferroelectricity is maintained. However, when the carrier concentration exceeds a critical threshold, the increased doping completely screens the ferroelectric polarization, leading to the loss of the fundamental characteristic of ferroelectric symmetry breaking[17]. Thus, achieving robust ferroelectricity against typical depolarization fields is essential. To achieve low-power devices and memory applications, exploring a method to make tin-based perovskite semiconductors ferroelectric is necessary.

Ferroelectric polarizations are majorly derived from nonferroelectric lattice distortions, charge ordering, or specific spin alignments. In addition, by incorporating polar molecular groups, the spatial arrangement of each group within hybrid materials will shift, creating the potential for the positive and negative charge centers to misalign[18]. Notably, the small-size imidazole molecule exhibits a significant dipole moment of 3.61 D aligned nearly parallel to the N-H bond[19]. As a derivative of the imidazole molecule, 2-methylbenzimidazole (MBI) introduces a methyl group that heightens the asymmetry of the molecular charge distribution, thereby enhancing its polarity and increasing the dipole moment[20]. By introducing MBI, a strong hydrogen bond forms with the H atom-containing group in the material whose orientation has been strengthened. Consequently, there exists significant potential for inducing ferroelectricity in Sn-based perovskite semiconductors.

In this work, Sn-based perovskite (93.3 mol% (FA$_{0.86}$Cs$_{0.14}$)SnI$_3$ and 6.7 mol% PEA$_2$SnI$_4$) semiconductor films are transformed into ferroelectric semiconductors, by introducing MBI molecule. The crystallinity of perovskite films is improved after MBI doping. Through the molecular reconfiguration of imidazole, the Sn-based perovskite exhibits both remarkable semiconductor properties and exceptional ferroelectricity, with the underlying physical mechanisms for this emergence having been elucidated. Due to the influence of ferroelectricity, the Sn-based perovskite transistors exhibit pronounced ferroelectric behavior, resulting in a notably low subthreshold swing (SS) value.

## Results
### Enhanced crystallinity of the perovskite semiconductor films doped with MBI molecule
The phase purity of recrystallized MBI was investigated by X-ray diffraction (XRD) (Fig. S1)[21]. The XRD results presented in Fig. 1a and Supplementary Fig. 2 indicate that the crystallinity of the perovskite films was enhanced following MBI doping. Fullwidth at half-maximum (FWHM) and peak position of (110) crystal plane in XRD of perovskite films with different MBI concentrations are provided in Supplementary Tables 1, 2. Compared with pristine, the FWHW of MBI-doped perovskite film decreases on (100), denoting crystallinity has been improved. All perovskite films exhibit three dominant peaks at around 14°, 28°, and 33°, corresponding to the (100), (200), and (122) crystal planes of the FASnI$_3$[15]. The doping of the MBI molecules enhances the absorption ability, while the absorption peak position remains unchanged, indicating that the introduction of MBI does not change the body structure (Fig. 1b). MBI was not incorporated into the crystal lattice. The primary material employed was a 2D/3D perovskite composite, with the 2D component constituting 6.7 mol% of the overall

material. XRD analysis revealed the characteristic crystal plane peaks associated with the 3D perovskite structure. The enhancement in absorption coefficient and the improved crystallinity demonstrates a significant improvement in the quality of the film following the introduction of MBI molecules. The rise in crystallinity further indicates a more orderly arrangement of the perovskite spatial structure. Analysis of the AFM images reveals that both films exhibit low roughness ($Rq$). Furthermore, the average grain size of the film incorporating 0.5 mol% MBI molecules is larger compared to the pristine films (Fig. 1c, d). After introducing MBI molecules, the crystallinity of grains increased, mainly reflected in the increase of grain size and longitudinal growth from the SEM images (Supplementary Fig. 3). We used grazing-incidence wide-angle X-ray scattering (GIWAXS) to scrutinize the orientation of the crystal structure further. As illustrated in Fig. 1e, f, while the pristine perovskite film displays broad Debye-Scherrer rings indicative of random orientation within the perovskite grains, the perovskite film doped with 0.5 mol% MBI exhibits distinct Bragg points, signifying a high degree of crystal orientation. The other GIWAXS results with various doping concentrations are presented in Supplementary Fig. 4, suggesting improved crystallinity after the introduction of MBI molecular. The time-resolved photoluminescence (TRPL) results reveal an increase in the average carrier lifetime of perovskite films with MBI, further underscoring the enhancement in film quality (Supplementary Fig. 5). This orderly arrangement lays the foundation for the emergence of ferroelectricity. Moreover, the improved film quality mitigates the defect-induced shielding effect on the ferroelectricity.

### Ferroelectricity validation of reconfigured perovskite semiconductors
Next, piezoelectric atomic force microscope (PFM) and switching spectroscopy PFM (SS-PFM) were used to discover evidence of ferroelectricity in the perovskite semiconductors incorporated with the MBI molecule. The thickness of all perovskite films is about 38 nm. As depicted in Fig. 2a, b, the PFM out-of-plane (OOP) phase and amplitude images of the pristine film did not exhibit a signal driven by alternating current (AC) bias. The local phase and amplitude curves of the pristine perovskite film are shown in Fig. 2c, which do not exhibit any ferroelectric characteristics. On the contrary, the OOP phase and amplitude image of perovskite films doped with 0.5 mol% MBI molecule exhibit distinct contrast, dedicating existing piezoelectric response. The OOP phase image reveals a distinctly clear boundary, while the OOP amplitude image exhibits amplitude variations exceeding 12 pm (Fig. 2d, e), which demonstrates that the film undergoes a 180-degree domain flip when driven by AC bias and exhibits contraction or expansion when driven by voltage. The well-defined hysteresis loops and "butterfly" curves compellingly demonstrate the robust ferroelectricity in doped 0.5 mol% MBI molecule perovskite films (Fig. 2f). The other PFM and SS-PFM results with various doping concentrations are presented in the Supplementary Fig. 6 and Fig. 7, suggesting again that the doping of MBI molecule can bring the ferroelectricity for perovskite semiconductors.

Second harmonic generation (SHG) spectroscopy is an efficient technique to ascertain spatial symmetry breaking. Compared to the pristine film, the perovskite film doped with 0.5 mol% MBI molecule exhibits an obvious frequency-doubled signal at 515 nm (Fig. 2g), confirming the spatial symmetry breaking. As we know, the remnant polarization intensity ($P_r$) is key parameters used to evaluate the ferroelectric properties of materials. However, obtaining parameters in ferroelectric semiconductors is challenging, as the leakage current in these materials can obscure the polarization switching current. To determine these values and further confirm the presence of ferroelectricity, a ferroelectric analyzer was employed to test the pristine and with MBI films, using metal-insulator-semiconductor-metal (MISM) structure to decrease leakage current. The polarization–electric field (P-E) hysteresis loop is shown in Fig. 2h, we can find a typical rectangular P-E

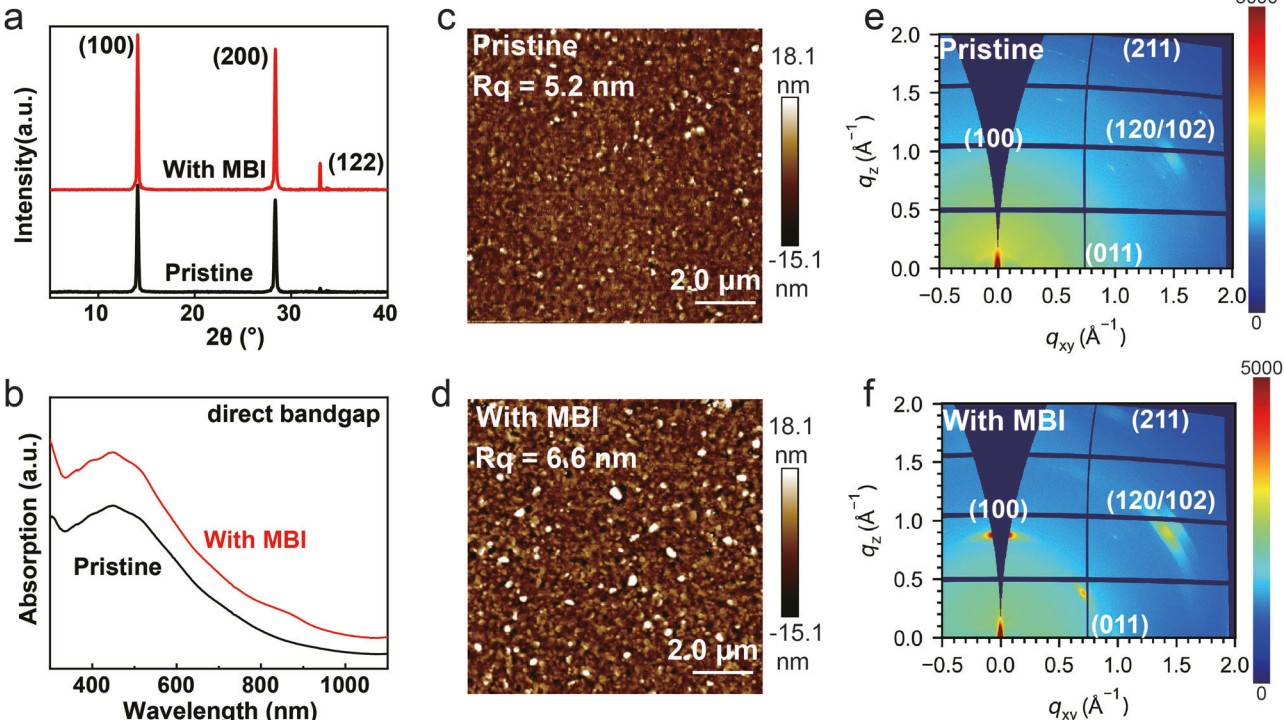

**Fig. 1 | Material characterizations of Sn-based perovskite films before and after doping with MBI molecule. a** XRD pattern, **b** absorption spectra (film thickness = 38 nm), **c**, **e** are the AFM and GIWAXS data for the pristine films. **d**, **f** are the AFM and GIWAXS data for the films doped with the MBI molecule.

ferroelectric loop, indicating obvious ferroelectric characteristics. The material is a ferroelectric semiconductor, which differs from traditional ferroelectric insulators due to its higher carrier concentration. This phenomenon arises from carrier movement during testing, resulting in a rounded P-V curve, rather than the typical rectangular shape seen in ferroelectric insulators. The P-V curves of the following ferroelectric semiconductors exhibit similar behavior to those in our study, such as SnS[16], β-CuGaO[8], $(KNbO_3)_{1-x}(BaNi_{1/2}Nb_{1/2}O_{3-\delta})_x$[22], and ST-3R $MoS_2$[23].

Obtained from the P-E loop data, the $P_r$ values of the perovskite film doped with 0.5 mol% MBI molecule is 23.2 μC/cm[2], respectively, which are the highest values among all the MBI doping concentrations. Supplementary Fig. 8b, e present the current-voltage (J-E) of the pristine and the doped with 0.5 mol% MBI molecule perovskite films, while the pristine does not exhibit polarization reversal current as the voltage changes, the perovskite films doped with MBI show polarization reversal current in the negative direction at about 20 V. The polarization reversal current remains with the range of voltage increasing, indicating the high stability of Sn-based ferroelectric perovskite semiconductors. Therefore, the robustness of ferroelectricity in ferroelectric semiconductors with high carrier concentration is proved. The other J-E and P-E results are presented in Supplementary Fig. 8c–h. Comparing the reported values from other ferroelectric semiconductors, such as SnS[16], β-CuGaO[8], α-$In_2Se_3$[24], $(KNbO_3)_{1-x}(BaNi_{1/2}Nb_{1/2}O_{3-\delta})_x$[22], and ST-3R $MoS_2$[23], the Sn-based perovskite semiconductor films prepared by MBI molecular reconfiguration achieve higher $P_r$ values, confirming its excellent ferroelectricity (Fig. 2i).

### Origin of ferroelectricity in Sn-based perovskite semiconductors doped with MBI molecule
The results discussed above highlight the reconfiguration effect of the MBI molecule, which effectively induces ferroelectricity in Sn-based perovskite semiconductor films. However, the physical mechanism of the emergence of ferroelectricity needs to be explored. Introducing the strongly electronegative MBI molecule, the terminal $NH_3^+$ group of the PEA molecule within 2D perovskite forms an intermolecular N-H…

N hydrogen bond with the MBI (Fig. 3a, b). The hydrogen bonding causes the PEA molecule to be twisted and rotated. Then, the ferroelectricity in Sn-based iodide films is induced by other nonpolar aberrations, where spontaneous polarization does not completely explain all the symmetry changes during the phase transition, corresponding to non-intrinsic ferroelectric properties. By incorporating the MBI polar molecular moiety to optimize structural distortion and employing a rational design of the polar framework, ferroelectricity subsequently emerges. The effectiveness of this transformation significantly displaces the positive and negative charge centers in the perovskite structure. This substantial displacement leads to a pronounced degree of charge center mismatch, thereby inducing strong ferroelectricity.

At the same time, the centers of positive and negative charges do not coincide, leading to the disappearance of central inversion symmetry and the splitting of the energy band. The energy band splitting in ferroelectric semiconductors induces ferroelectricity by resulting in two spin-polarization bands, known as the Rashba effect[25]. This method of generating ferroelectricity is validated through density functional theory (DFT) calculations, which offer a robust certification for predicting and confirming ferroelectricity[26-28]. This effect will be experimentally verified in the next section.

We further studied material physics using X-ray photoelectron spectroscopy (XPS). Compared to the pristine film, the N 1s peak of perovskite film doped with 0.5 mol% MBI molecule shifts toward lower binding energies, indicating the presence of intermolecular interactions. Specifically, the highly negatively charged N atom in MBI interacts with the N-H group in PEA, resulting in a C-N=C binding energy offset exceeding 0.9 eV. Moreover, the N-H content increased from 8.76 to 35.91% (Fig. 4a, b), providing strong evidence that the source of the ferroelectricity is the MBI molecule-induced hydrogen-bonding reconfiguration. Moreover, the XPS results from other Sn-based perovskite films with different MBI concentrations also show a notable increase in the N-H ratio, compared to the pristine films (Supplementary Fig. 9).

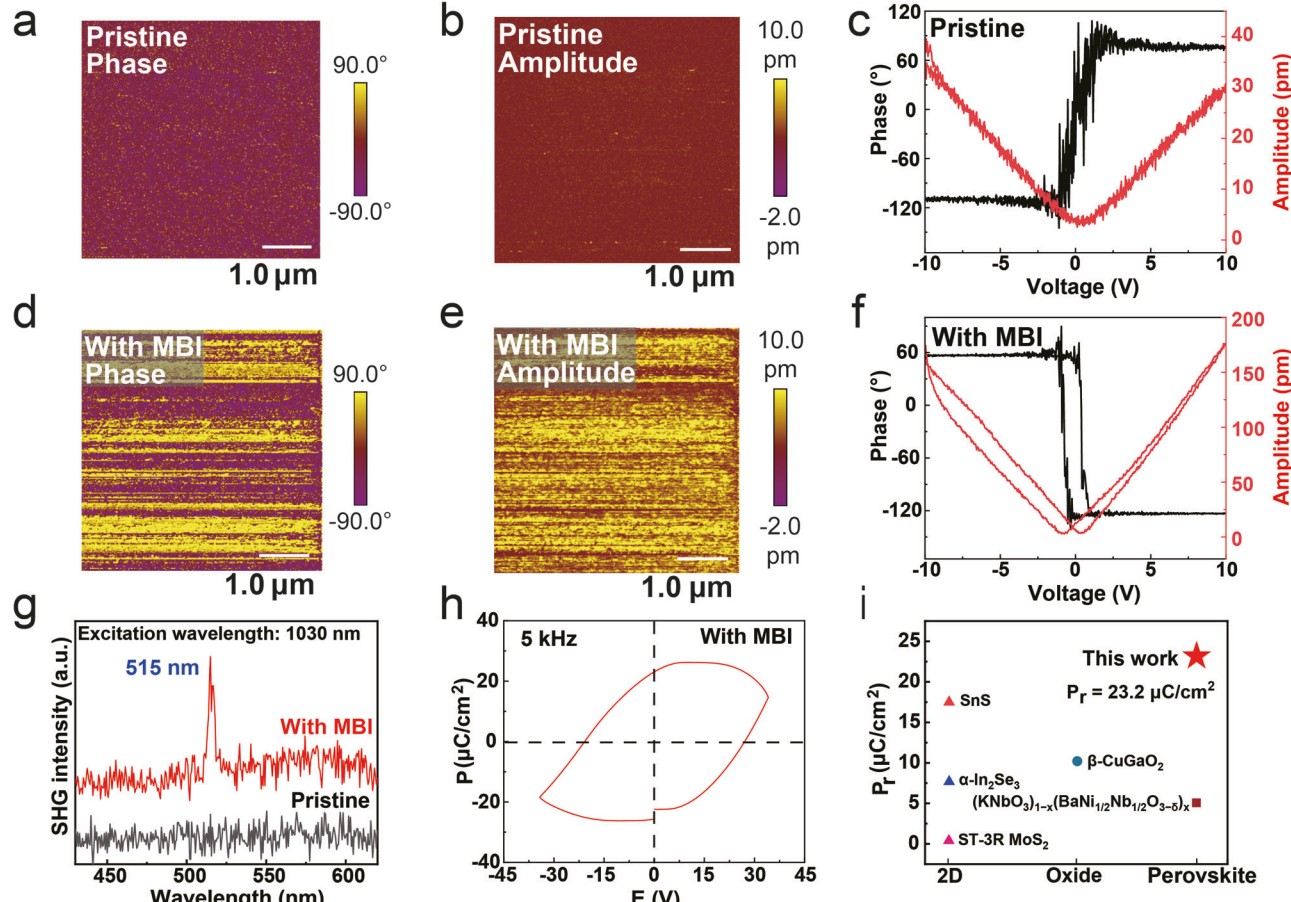

**Fig. 2 | PFM, SHG, and the ferroelectric properties of Sn-based perovskite films.** **a**, **b** are the PFM phase and amplitude for the pristine films. **d** and **e** are the PFM phase and amplitude for the films doped with MBI molecule. **c**, **f** are SS-PFM data for the pristine and the doped films, **g** SHG spectra of the pristine and doped films.

**h** Polarization-electric-field loops measured with a metal-insulator-semiconductor-metal capacitor for doped films. **i** Our $P_r$ value compared to the other ferroelectric semiconductors. Piezoelectric hysteresis loops of PFM were conducted by applying direct current (DC) bias that sweeps from −10.0 to 10.0 V.

Photoluminescence spectroscopy (PL) and circular polarization luminescence spectroscopy (CPL) tests were carried out to verify the Rashba effect, which enables their selective excitation by circularly polarized light. With the doping of 0.5 mol% MBI molecule, the PL curve of the perovskite film bifurcates into two peaks, compared to the pristine film. The high-energy peak at 1.493 eV corresponds to the free exciton peak, while the low-energy peak at 1.483 eV is attributed to the indirect transition due to Rashba splitting[29] (Fig. 4c). This bifurcation aligns with the disruption of spatial symmetry in the perovskite film, as indicated by the generation of the second harmonic generation (SHG) signal (Fig. 2g). The degree of splitting in the PL peak and the SHG signal varies with the introduction of different MBI concentrations, as illustrated in the Supplementary Fig. 10a, b. The CPL spectroscopy measurements were conducted on both pristine and MBI-doped films to determine the degree of spontaneous energy band splitting. This was achieved by analyzing the disparities in fluorescence spectra excited by left-handed and right-handed circularly polarized light[29]. Figure 4d demonstrates that MBI doping increased the left-right fluorescence intensity difference at the 1.493 eV emission peak from −16.5 to −20. This enhancement indicates a greater degree of self-selective cleavage, further confirming the Rashba effect.

## High-performance transistors based on the ferroelectric perovskite semiconductors

Based on the Sn-based ferroelectric perovskite semiconductor, ferroelectric field-effect transistors (FeFETs) with bottom-gate top-contact structure were fabricated (Fig. 5a). From the transfer curves of pristine transistors, the backward current is lower than the forward current, exhibiting typical interface defects and bulk defects capture carrier characteristics (Fig. 5b). However, due to the effects of polarization-electric field, the backward current exceeds the forward current, indicating pronounced ferroelectric hysteresis for the Sn-based perovskite FeFETs. In addition, the output curves in Supplementary Fig. 11 further suggest that the FeFETs have good contact behaviors, and the contact resistance of the transistor with MBI was measured and determined to be 347 Ω·cm (Supplementary Fig. 12).

We conducted further measurements of the transfer curves across varying scan speeds, scan ranges, $V_{DS}$, and temperatures. The scanning window remained largely consistent, reaffirming that the hysteresis originates from the ferroelectricity of FeFETs. More detailed explanations are provided in Supplementary Fig. 13a–e. Moreover, no significant change can be observed in the hysteresis window at varying temperatures within the subthreshold region (Fig. 5c and Supplementary Fig. 13f), thus eliminating the interference from ion migration. From a device-level perspective, the perovskite transistor doped with 0.5 mol% MBI molecule is indeed a FeFET, whose average carrier lifetime of the perovskite layer reaches the highest point compared with other doping ratios (Supplementary Fig. 5). Furthermore, with increasing the MBI concentrations, the hysteresis curves transition from defect-dominated to ferroelectric-dominated, and eventually defect-dominated (Supplementary Fig. 14). Meanwhile, TRPL showed the average carrier lifetime initially rises and subsequently declines,

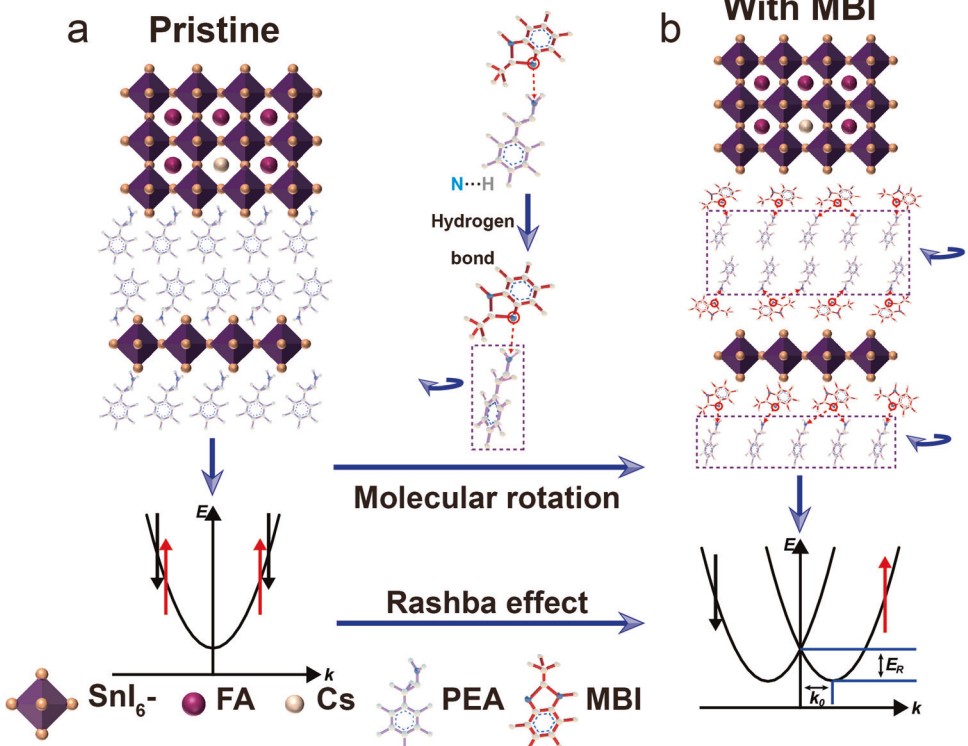

**Fig. 3 | Schematic diagram of ferroelectric generation mechanism. a** Schematic diagram of the structure and energy level in pristine Sn-based perovskite semi-conductor. **b** Schematic diagram of the structure and energy level in perovskite ferroelectric semiconductor after doping with MBI molecule.

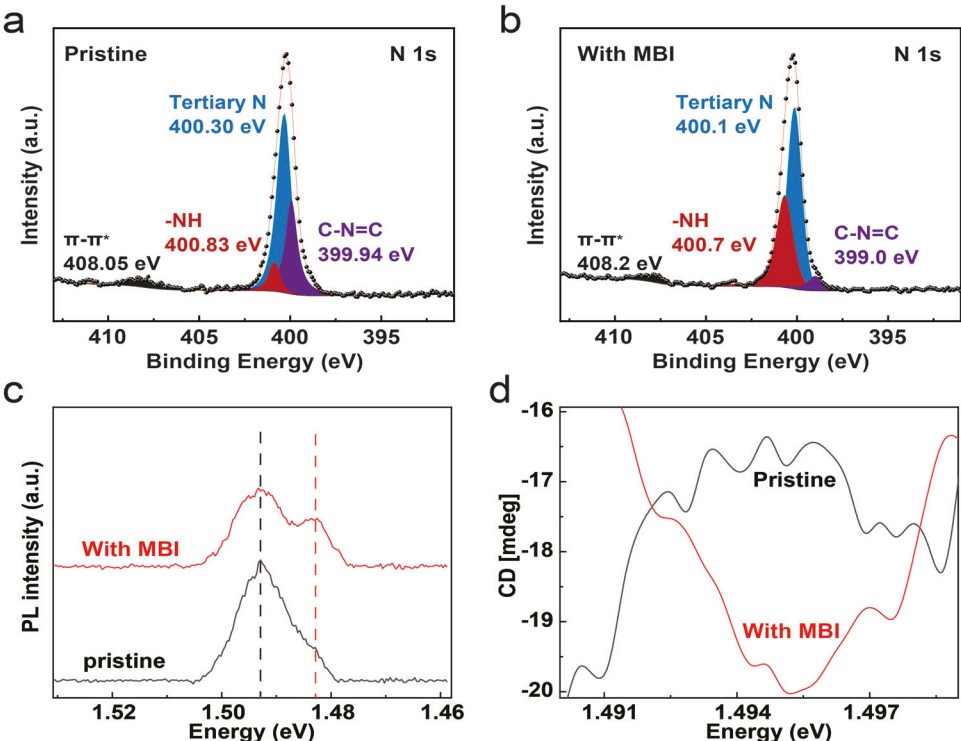

**Fig. 4 | PL, CPL, and XPS spectra of the Sn-based perovskite semiconductor films. a, b** are the X-ray photoelectron spectroscopy of pristine and doped films. **c** photoluminescence spectra, **d** circular polarization luminescence spectra.

mirroring the trend in film quality (Supplementary Fig. 5). Due to higher doping concentrations, the substantial increase in bulk defects generates a defect field that ultimately overwhelms the ferroelectric field. The predominant bulk defect is a PEA vacancy, which may passivate Sn vacancies[30–32], allowing for the assessment of PEA incorporation by examining Sn vacancy behavior. XPS results reveal that with increasing MBI incorporation, the concentration of divalent tin vacancies initially rises and then declines (Supplementary Fig. 15).

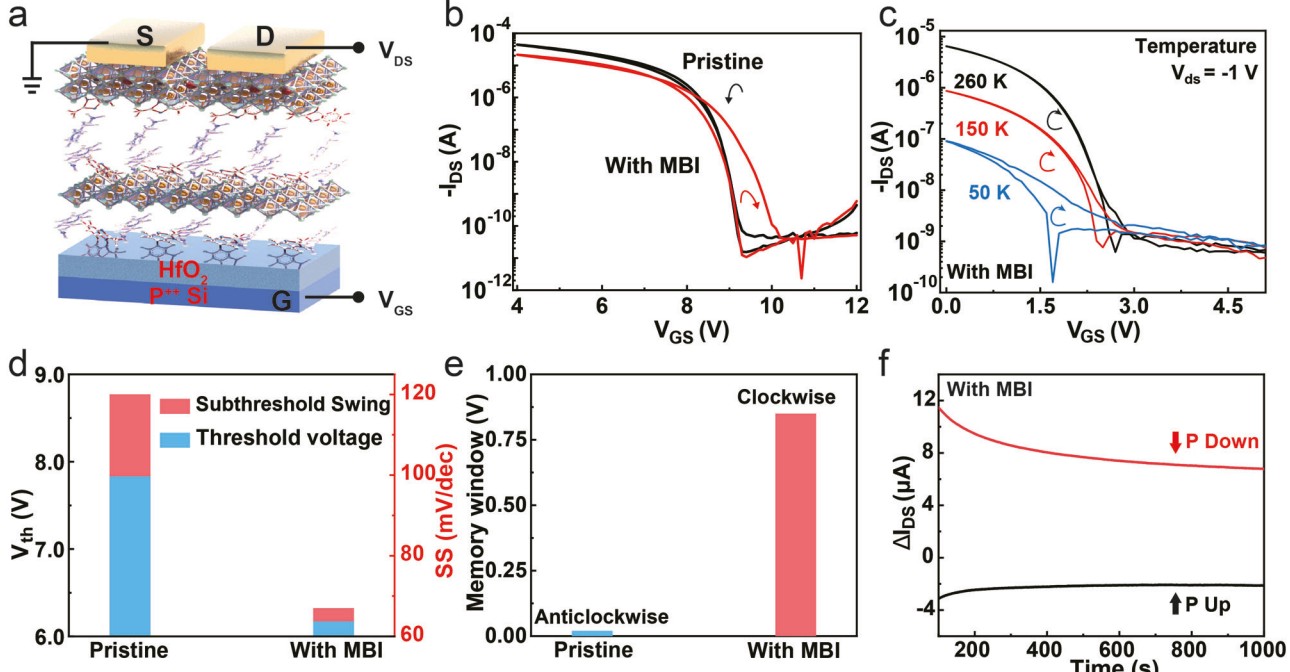

**Fig. 5 | Electrical characteristics of Sn-based perovskite transistors. a** Schematic diagram of the back-gated transistor structure. **b** Transfer curves of pristine and doped transistors. **c** Transfer curves of Sn-based perovskite FeFETs measured at different temperatures. **d, e** Comparison of the $V_{th}$, $SS$, and memory window values of the pristine and doped transistors. **f** Memory performance of the Sn-based perovskite FeFETs.

Excessive MBI introduction inhibits PEA incorporation into the two-dimensional structure, leading to the formation of PEA vacancies. The reduction of Sn vacancy reduces the carrier concentration and is conducive to the generation of ferroelectricity.

To further investigate the effects of ferroelectricity, we calculated the key transistor performance parameters, including threshold voltage ($V_{th}$) and $SS$ values. After doping with MBI, the $V_{th}$ decreased from 7.83 to 6.17 V, while the SS decreased from 120 to 67 mV/dec, which extraction method was provided in Supplementary Fig. 16[33,34], as given in Fig. 5d, exhibiting the potential for low-voltage operation and low-power consumption devices. Remarkably, the SS value is lower than that observed from two-dimensional $\alpha$-In$_2$Se$_3$ FeFETs[7] (650 mV/dec) and DPPT-TT polymer transistors (85 mV/dec)[35], and even approaches the theoretical diffusion limit of 60 mV/dec. This is attributed to the strong ferroelectric field after MBI doping. The polar charge then contributes to gate control and also diminishes the impact of carrier scattering, resulting the high-performance FeFETs[11]. Compared to the pristine FET, the hysteresis window increases from 0.02 to 0.85 V after MBI doping (Fig. 5e), suggesting an enhanced memory ability. Finally, the memory performance was evaluated by applying 50 write voltages (10 V and 1 s) in opposite directions (Fig. 5f). Both the upward and downward polarization states were sustained for over 1000 s, conclusively proving the memory capability of the ferroelectric transistors.

## Discussion

We prepared Sn-based perovskite semiconductor films with strong room-temperature ferroelectricity, by introducing a strongly polar molecule group (MBI) to achieve intermolecular hydrogen bonding reconfiguration. The XRD results indicate that the film crystallinity increases after MBI doping. Besides high-performance semiconducting properties, the optimized perovskite films also exhibit robust ferroelectricity with high remanent polarization. The ferroelectricity originates from the noncoincidence of positive and negative charge centers, due to the spatial symmetry breaking by MBI doping. Utilizing the advantages of strong ferroelectric properties, high-performance perovskite ferroelectric semiconductor transistors with ultra-low $SS$ value were fabricated. This work offers a Sn-based perovskite ferroelectric semiconductor film and gives a deep insight into understanding ferroelectric material physics.

## Method
### Preparation of perovskite films

PEAI (1.6 M was dissolved in DMF), FAI (0.8 M was dissolved in DMF), CsI (0.8 M was dissolved in DMSO), SnI$_2$ (0.8 M was dissolved in DMF, SnF$_2$ (0.08 M) was added), MBI (0.01 M was dissolved in DMSO) were used as the mother solution and heated at 60 °C overnight. Perovskite precursors (0.2 M, DMF:DMSO volume ratio 3:1), 93.3 mol% (FA$_{0.86}$Cs$_{0.14}$)SnI$_3$ and 6.7 mol% PEA$_2$SnI$_4$ with different mol ratio (0, 0.1, 0.3, 0.5, 2.0, and 4.3) of MBI molecule were prepared by mixing the mother solutions. The precursor solution was continuously stirred for 2 h. All processes were performed in a nitrogen-filled glove box. The perovskite precursor was spin-coated onto the substrate at 5000 rpm for 60 s, and 100 μL of anti-solvent chlorobenzene was dropped onto the substrate for 10 s after the start of spin-coating. Then the films were thermal annealing at 100 °C for 5 min. The film thickness measured by the step profiler (Zeptools JS100A) is 38 nm.

### Perovskite film characterizations

The morphology and thickness of the films were measured by atomic force microscope (AFM, Bruker Dimension Edge). The film thickness was also measured by the step profiler (Zeptools JS100A). Piezoelectric atomic force microscope (PFM) images of the film were obtained by Bruker Dimension Icon. The crystal structure of the perovskite films was measured by XRD (Bruker D8 Advance). The ultraviolet-visible absorption spectroscopy experiments and photoluminescence spectroscopy (PL) were conducted in Spectrum-Shanghai SP-1920 and HITACHI F-4600 with an excitation wavelength of 550 nm, respectively. The second harmonic generation (SHG) spectra were measured by a 1030 nm fs pulsed laser (Idea Optics). The circular polarization

luminescence (CPL) was tested by CPL-300, Jasco. The X-ray photo-electron spectroscopy (XPS) was performed by AXIS Kratos Supra⁺. The P-V loop and I-V loop were tested with a double-wave method by Precision Premier II Ferroelectric Tester. The scanning electron microscope (SEM) was performed by JEOL JSM-6701F. The time-resolved photoluminescence (TRPL) was performed by the Horiba Jobin Yvon system.

### Device fabrications and measurements

The 50 nm $HfO_2$ dielectric layer was grown by atomic layer deposition (ALD) on a highly doped p-type silicon substrate, then the substrate was treated with ultraviolet ozone for 20 min. After preparing the perovskite films, 50 nm gold was thermally evaporated as the source and drain electrodes. The channel length and width are 1200 and 150 μm, respectively. The electrical properties of the perovskite transistors were measured by a semiconductor parameter analyzer (FS-Pro and B1500) equipped with a liquid helium probe station (Lakeshore CRX 6.5 K).

### Reporting summary

Further information on research design is available in the Nature Portfolio Reporting Summary linked to this article.

## Data availability

The data that support the findings within this paper and other findings of this study are available from the corresponding authors upon reasonable request.

## Code availability

There is no code involved in the research work of this paper.

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

## Acknowledgements

This work was supported by National Natural Science Foundation of China (Grant Nos. 62374043, 62204055, and 62304043), Shanghai Pujiang Program (Grant No. 2022PJD004), Shanghai Oriental Talent Program Youth Project (2022), and China Postdoctoral Science Foundation (Grant Nos. 2022M720751, 2023T160119, and 2022M720748).

## Author contributions

W. L. conceived and supervised the research. Y.L., S.Y., and W. L. designed the project, performed the experiments, and collected the data. S.Y., X.Y., and Y.W. assisted in the preparation of perovskite thin films, L.H., L.Z., and Z.A. assisted in the test of the ferroelectric analysis tester, E.L. and J.W. assisted in the preparation of the $HfO_2$ dielectric layer, Y.Z. assisted in the PFM test, and Y.Y. assisted in the GIWAX test. W.L. and J.C. conceived the idea. All authors wrote this manuscript and contributed to reviewing the paper.

## Competing interests

The authors declare no competing interests.
