## [Transparent Peer Review file · Nature Communications]

Emergence of ferroelectricity in Sn-based perovskite semiconductor films by iminazole molecular reconfiguration

Corresponding Author: Professor Wenwu Li

Version 0:

Reviewer comments:

Reviewer #1

(Remarks to the Author)

Ferroelectric semiconductors have the advantages of switchable polarization ferroelectric field regulation and semiconductor transport characteristics, which are promising applications in high-performance electronics and optoelectronic devices, and in-sensor computing. However, it is difficult to develop a film with both robust ferroelectricity and excellent semiconductor properties. In this work, Liu et al used imidazole molecules to dope the Sn-based perovskite semiconductor films. Owing to molecular reconfiguration effects, the Sn-based perovskite semiconductors transform into ferroelectric semiconductors. The topic of perovskite semiconductors has been popular in recent years, and the phenomena observed in this paper are interesting. So I would like to suggest a minor revision for this manuscript. Before publication, some questions should be addressed.

1. From Fig.1, the absorption coefficient of the films is enhanced, and the crystal quality of the films is improved, after MBI ferroelectric molecular doping. Is there any relationship between the absorption coefficient, crystallinity, and ferroelectricity of the Sn-based perovskite semiconductor films?
2. The ferroelectricity of molecular-based perovskite films is even higher than that of Sn-based perovskite ferroelectric semiconductors. The authors should explain the advantages of this work.
3. The authors claim that after doping with MBI molecules, the ferroelectricity of Sn-based perovskite semiconductors, such as Pr and Ec (Fig. 2i), is higher than other ferroelectric semiconductors. However, the polarization–electric-field loops in Fig. 2h are rounded and they may reflect a weak ferroelectricity, compared to traditional perovskite ferroelectric materials (such as BiFeO₃). The authors should address this point.
4. In the abstract and introduction section, the advantages of ferroelectric semiconductors for transistor applications are presented. However, the advantages of Sn-based ferroelectric semiconductor transistors are not significant in Fig. 5, compared to the undoped semiconductor transistors. I suggest the authors carry out more device experiments, to find the enhancement of electrical parameters for ferroelectric transistors.
5. Some issues with the English in the manuscript should be revised. Some of the figures are not clear, and some fonts of the diagrams are not readable.

Reviewer #2

(Remarks to the Author)

This manuscript reports ferroelectric and semiconductor properties of Sn-based perovskites. By doping with imidazole-based molecule (MBI), Liu et al discovered emergence of ferroelectricity with high remanent polarization and low coercive field. The ferroelectricity is stable under bias voltage, and the origin rises from symmetry breaking by hydrogen bonding between phenylethylammonium and nitrogen of MBI. The ferroelectric transistors demonstrate low subthreshold swing of 80 mV/dec. It is our opinion that the discovery of ferroelectric behavior in Sn-based perovskites, which were applied to various high-performance field-effect transistors is very interesting, and the presented ferroelectric data are quite convincing, but the mechanism requires slightly more analysis. I therefore consider this manuscript as suitable for publication in Nature Communications, only after provided an extensive reply to the following comments and inquires.

1. The authors state that it is difficult to develop a thin-film with robust ferroelectricity and excellent semiconductor properties

in the first paragraph of the introduction. In the second paragraph of the introduction, the authors state that in Sn-based perovskite semiconductors, the high carrier concentration causes the internal electrical field to be not effectively screened, weakening ferroelectric polarization. The authors should give more detailed reason behind this phenomenon with supporting references, which eventually links to the success of achieving both in this manuscript. Also, the high carrier concentration of Sn-based perovskite semiconductors is often reduced by engineering of the Sn vacancy. Please comment on the efficiency of this method for arising ferroelectricity.

2. The authors demonstrate an increase in crystallinity after MBI doping through increase in peak intensity of XRD. While this applies for Fig. 1, Supplementary Fig. 2 shows increased intensity for certain doping percentages of MBI: 0.3, 0.5 4.3 mol%. Is there a specific reason behind this trend? Also, PL and SHG intensities in Supplementary Fig. 7 show a comparable trend of intensities according to dopant molar percentage with each other, however, the trend seems to differ with XRD. XRD and PL intensity often show similar relationships as they are related to film quality and crystallinity. Please comment on this difference.

3. The AFM images of pristine and MBI doped thin-films are discussed with increase in film crystallinity. Through AFM images, the thin-film quality looks similar. The authors could comment further on this, provided with SEM images of the two thin-films.

4. According to the mechanism presented by the authors, the ferroelectricity arises from the breaking of symmetry with MBI doping. This causes compression of the space volume and the hydrogen bonding causing PEA molecule to be twisted and rotated. This phenomenon is well supported by XPS data, however, it seems to differ with XRD data. Lattice contortion, compression or expansion can cause peak shifts in XRD measurements. Please comment on this difference.

5. While the degree of absorption can indeed vary due to the absorptive properties of the thin film, it is also a parameter that can change significantly with the thickness of the film. Therefore, when claiming an increase in absorption, it is imperative to specify the thickness of the thin film as well.

6. Could the authors provide GIWAXS data for thin films with different amounts of MBI? Since there is no clear trend in the crystallinity based on the XRD data, it is suggested to include GIWAXS data for each amount of MBI in the supplementary information to support the claims of improved crystallinity.

7. In Supplementary Figure 5, P-V test data for the pristine sample is missing. To clearly demonstrate the effect of MBI on the induction of ferroelectric properties, please include data for the pristine samples as well.

8. According to the mechanism proposed by the authors, MBI and PEA interact strongly via hydrogen bonds, causing PEA to adopt a twisted and rotated structure. Based on this, I would expect MBI to inhibit the incorporation of PEA into the perovskite lattice, potentially increasing the number of defects at grain boundaries or surfaces. This trend of increased defects can be inferred from the transfer characteristics shown in Supplementary Figure 10, which vary with the MBI content. How significant do you consider the contribution of these defects to the hysteresis observed in the transfer curves? If you claim that the contribution of charge carrier trapping at these defects is minimal, please substantiate this with time-resolved photoluminescence (TRPL) or further electrical characterization that demonstrates minimal impact from trapping. Additionally, the schematic diagram illustrating the structure shows PEA's benzene ring oriented towards the perovskite lattice, which is likely incorrect since NH_3^+ typically binds to the A cation site of the perovskite. Please correct this in the diagram.

9. In Figure 4, the x-axis of the PL data appears to represent energy in eV, but it is incorrectly labeled as wavelength. Please check and correct this notation.

Additionally, Supplementary Figure 7 shows that the introduction of MBI results in the formation of a shoulder peak in the PL results due to Rashba splitting. However, the intensity of this shoulder peak does not exhibit a clear trend with increasing amounts of MBI. Could you explain why?

10. Based on the results from the output curve measurements, it has been concluded that the devices show good contact behavior. Please try using techniques such as the transmission line method (TLM) to accurately evaluate the contact resistance.

11. The authors suggested that an excess amount of MBI would lead to a significant increase in bulk defects. Could you specify what types of bulk defects are anticipated?

12. How did the authors determine the V_{th} and the SS? While Figure 5b shows minimal changes in SS and V_{th} , the manuscript indicates a substantial difference.

If extrapolation was used, please show how it was derived from the graph. If another method was employed, please also describe that method.

13. The transfer curves measured at low temperatures display only hysteresis in the subthreshold region, while virtually no hysteresis is observed in other regions. This makes it challenging to assert that there is no change in the hysteresis window with temperature. Please check and correct this.

Additionally, why do you think the hysteresis observed in the transfer curves at low temperatures is significantly smaller than that at room temperature?

Furthermore, according to Supplementary Figure 9f, the transfer curve of a transistor containing 0.5 mol% measured near room temperature at 290K shows considerable differences in hysteresis compared to what is displayed in Figure 5b. What is the reason?

Reviewer #3

(Remarks to the Author)

Reviewer #4

(Remarks to the Author)

Ferroelectric semiconductors offer the dual advantages of switchable polarization and semiconductor transport characteristics, making them highly promising for applications in ferroelectric transistors and nonvolatile memory devices. However, preparing Sn-based perovskite films with robust ferroelectric and semiconductor properties remains challenging. In this study, authors have shown that doping Sn-based perovskite films with 2-methylbenzimidazole (MBI) effectively transforms them into ferroelectric semiconductor films through molecular reconfiguration. These reconfigured films exhibit a high remanent polarization (P_r) of $23.2 \mu\text{C}/\text{cm}^2$ and a low coercive field (E_c) of $7.2 \text{ kV}/\text{cm}$. The emergence of ferroelectricity is attributed to the enhancement of hydrogen bonds following imidazole molecular doping, which breaks spatial symmetry and causes the positive and negative charge centers to become non-coincident. Notably, transistors based on these perovskite ferroelectric semiconductors demonstrate a low subthreshold swing of $80 \text{ mV}/\text{dec}$, further highlighting the advantages of introducing ferroelectricity.

This work presents a novel approach to realizing Sn-based ferroelectric semiconductor films for electronic device applications. The manuscript is well-written, short, and concise, with no ambiguity. I recommend it for publication in Nature Communications.

One minor comment:

The origin of ferroelectricity is not discussed properly. I suggest the authors revise the discussion of ferroelectric origin with further DFT calculations and/or by citing previous DFT-based published papers.

Version 1:

Reviewer comments:

Reviewer #1

(Remarks to the Author)

The authors have addressed all of my previous concerns. I think the manuscript is now suitable for publication after a careful proofreading.

Reviewer #2

(Remarks to the Author)

The paper is properly revised and I suggest the paper is accepted as it is.

Reviewer #3

(Remarks to the Author)

Reviewer #4

(Remarks to the Author)

I have seen the response to my previous comments. I recommend the present version for publication.

Responses to Reviewer #1

CI: Ferroelectric semiconductors have the advantages of switchable polarization ferroelectric field regulation and semiconductor transport characteristics, which are promising applications in high-performance electronics and optoelectronic devices, and in-sensor computing. However, it is difficult to develop a film with both robust ferroelectricity and excellent semiconductor properties. In this work, Liu et al used imidazole molecules to dope the Sn-based perovskite semiconductor films. Owing to molecular reconfiguration effects, the Sn-based perovskite semiconductors transform into ferroelectric semiconductors. The topic of perovskite semiconductors has been popular in recent years, and the phenomena observed in this paper are interesting. So I would like to suggest a minor revision for this manuscript. Before publication, some questions should be addressed.

R1: We greatly appreciate the reviewer for the valuable and constructive comments regarding our manuscript. We have carefully revised the manuscript based on your suggestions. Our detailed response to each comment is provided below.

CI.1. From Fig.1, the absorption coefficient of the films is enhanced, and the crystal quality of the films is improved, after MBI ferroelectric molecular doping. Is there any relationship between the absorption coefficient, crystallinity, and ferroelectricity of the Sn-based perovskite semiconductor films?

R1.1: Yes, in this work, there is a relationship between film absorption, crystallinity, and ferroelectricity. The enhancement in absorption coefficient and the improved crystallinity demonstrate a significant improvement in the quality of the film following the introduction of MBI molecules. The rise in crystallinity further indicates a more orderly arrangement of the perovskite spatial structure. This orderly arrangement lays the foundation for the emergence of ferroelectricity. Moreover, the improved film quality mitigates the defect-induced shielding effect on the ferroelectricity. The time-resolved photoluminescence (TRPL) results reveal an increase in the average carrier lifetime of perovskite films with MBI compared with pristine, further underscoring the enhancement in film quality. The supplementary explanation has been added to the first part of the manuscript result, and the TRPL diagram has been added to Supplementary Figure 5.

Fig.R1. TRPL results of the Sn-based perovskite films with different doping concentrations of MBI molecule.

Revision: In the revised manuscript, we have added the sentences ‘The enhancement in absorption coefficient and the improved crystallinity demonstrates a significant improvement in the quality of the film following the introduction of MBI molecules. The rise in crystallinity

further indicates a more orderly arrangement of the perovskite spatial structure.”, page 5, line 15, in the part of “Enhanced crystallinity of the perovskite semiconductor films doped with MBI molecule” to clarify the relationship between the absorption coefficient, crystallinity, and ferroelectricity of the Sn-based perovskite semiconductor films. Meanwhile, we also have added Supplementary Fig. 5 and the sentence ‘The time-resolved photoluminescence (TRPL) results reveal an increase in the average carrier lifetime of perovskite films with MBI, further underscoring the enhancement in film quality (Supplementary Fig. 5). This orderly arrangement lays the foundation for the emergence of ferroelectricity. Moreover, the improved film quality mitigates the defect-induced shielding effect on the ferroelectricity.’, page 6, line 7, to explain the enhanced film quality with MBI.

CI.2. The ferroelectricity of molecular-based perovskite films is even higher than that of Sn-based perovskite ferroelectric semiconductors. The authors should explain the advantages of this work.

R1.2: The advantages of our work lie in the ability to transform perovskite semiconductors with high carrier concentrations into ferroelectric perovskite semiconductors through molecular reconfiguration. This approach yields perovskite with both exceptional semiconductor and ferroelectric properties. Molecular-based perovskite film of this work belongs to hybrid improper ferroelectric (HIF) materials, whose polarization can arise through trilinear coupling with two nonpolar structural distortions [*Phys. Rev. Lett.* **106**, 107204 (2011), *Journal of Solid State Chemistry* **195**, 11–20 (2012)], allowing for the rational design of polar structures by optimizing ubiquitous structural distortions. In particular, the perovskite used is RP-type perovskite, whose layered structure is more conducive to the generation of ferroelectricity trilinear coupling with structural distortions. The effectiveness of this transformation is rooted in the fact that MBI molecules possess large dipole moments, rotating the PEA with hydrogen bonds, which significantly displace the positive and negative charge centers in the perovskite structure. This substantial displacement leads to a pronounced degree of charge center mismatch, thereby inducing strong ferroelectricity.

Revision: In the revised manuscript, we have added the sentence ‘The effectiveness of this transformation significantly displaces the positive and negative charge centers in the perovskite structure. This substantial displacement leads to a pronounced degree of charge center mismatch, thereby inducing strong ferroelectricity.’, page 9, line 5, in the part of “Origin of ferroelectricity in Sn-based perovskite semiconductors doped with MBI molecule” to explain the advantages of this work, demonstrating the reason why the ferroelectricity of molecular-based perovskite films is even higher than that of Sn-based perovskite ferroelectric semiconductors.

CI.3. The authors claim that after doping with MBI molecules, the ferroelectricity of Sn-based perovskite semiconductors, such as P_r and E_c (Fig. 2i), is higher than other ferroelectric semiconductors. However, the polarization–electric-field loops in Fig. 2h are rounded and they may reflect a weak ferroelectricity, compared to traditional perovskite ferroelectric materials (such as BiFeO_3). The authors should address this point.

R1.3: Thank you for your suggestion. We have incorporated discussion into this point. The material is a ferroelectric semiconductor, which differs from traditional ferroelectric insulators due to its higher carrier concentration. This phenomenon arises from carrier movement during testing, resulting in a rounded P-V curve, rather than the typical rectangular shape seen in ferroelectric insulators. The P-V curves of the following ferroelectric semiconductors exhibit similar behavior to those in our study, such as $\beta\text{-CuGaO}_2$ [*Rare Met.* **41(3)**, 972–981 (2022)], $(\text{KNbO}_3)_{1-x}(\text{BaNi}_{1/2}\text{Nb}_{1/2}\text{O}_{3-\delta})_x$ [*Chem. Mater.* **34**, 4274–4285 (2022)], shear-transformed 3R-MoS_2 [*Nature Electronics* **7**, 29–38 (2024)], SnS [*ACS Nano* **14**, 7628–7638 (2020)]. Supplementary explanation has been added to the second part of the manuscript result.

[Figure redacted]

Fig.R2. P-V loop of β -CuGaO₂ [*Rare Met.* **41(3)**, 972–981 (2022)].

[Figure redacted]

Fig.R3. P-V loop of (KNbO₃)_{1-x}(BaNi_{1/2}Nb_{1/2}O_{3- δ})_x [*Chem. Mater.* **34**, 4274–4285 (2022)].

[Figure redacted]

Fig.R4. P-V loop of shear-transformed 3R-MoS₂ [*Nature Electronics* **7**, 29-38 (2024)].

[Figure redacted]

Fig.R5. P-V loop (the red line) of SnS [ACS Nano 14, 7628–7638 (2020)].

Revision: In the revised manuscript, we have added the following sentence ‘The material is a ferroelectric semiconductor, which differs from traditional ferroelectric insulators due to its higher carrier concentration. This phenomenon arises from carrier movement during testing, resulting in rounded P-V curve, rather than the typical rectangular shape seen in ferroelectric insulators. The P-V curves of the following ferroelectric semiconductors exhibit similar behavior to those in our study, such as SnS, β -CuGaO, $(\text{KNbO}_3)_{1-x}(\text{BaNi}_{1/2}\text{Nb}_{1/2}\text{O}_{3-\delta})_x$, and ST-3R MoS₂.’, page 7, line 17, to the part “Ferroelectricity validation of reconfigured perovskite semiconductors” for explain this point.

CI.4. In the abstract and introduction section, the advantages of ferroelectric semiconductors for transistor applications are presented. However, the advantages of Sn-based ferroelectric semiconductor transistors are not significant in Fig. 5, compared to the undoped semiconductor transistors. I suggest the authors carry out more device experiments, to find the enhancement of electrical parameters for ferroelectric transistors.

R1.4: Thanks for your valuable suggestion. We carried out more device experiments, fabricating transistor with SS as low as 67 mV/dec. Compared with pristine transistor, the SS of ferroelectric transistor is reduced by one time, approaching the theoretical diffusion limit of 60mV/dec. Compared with pristine, the SS of the Sn-based ferroelectric semiconductor transistors decreased from 120 mV/dec to 67 mV/dec, while the V_{th} reduced from 7.83 V to 6.17 V, and the hysteresis window expanded from 0.02 V to 0.85 V. These results clearly demonstrate the enhancement effect of the ferroelectric field on the electrical parameters of transistor devices.

Fig.R6. The optimized transfer curve of Sn-based ferroelectric semiconductor transistors.

Fig.R7. Electrical characteristics of Sn-based perovskite transistors.

Revision: In the revised manuscript, we have revised the '5.72 V' to '6.16 V', '80 mV/dec' to '67 mV/dec', page 12, line 5, in the section of "High-performance transistors based on the ferroelectric perovskite semiconductors". Meanwhile, the manuscript Figure 5 and Supplementary Figure 14 have been updated.

CI.5. Some issues with the English in the manuscript should be revised. Some of the figures are not clear, and some fonts of the diagrams are not readable.

R1.5: Thank you for your suggestion. We have revised the manuscript, including the text, figures and fonts in the diagrams, etc.

Revision: In the revised manuscript, we have polished the sentence, such as:

"Effects from the ferroelectricity, the Sn-based perovskite transistors show obvious ferroelectric behavior, leading to a low subthreshold swing (SS) value." to "Due to the influence of ferroelectricity, the Sn-based perovskite transistors exhibit pronounced ferroelectric behavior, resulting in a notably low subthreshold swing (SS) value.", page 5, line 2.

"The aforementioned results demonstrate the reconfiguration effect of the MBI molecule, which successfully induced ferroelectricity in Sn-based perovskite semiconductor films." to "The results discussed above highlight the reconfiguration effect of the MBI molecule, which effectively induces ferroelectricity in Sn-based perovskite semiconductor films.", page 8, line 16.

"We further measured the transfer curves at different scanning speeds, scan ranges, V_{DS} , and temperatures. The scanning window exhibits almost unchanged trends, confirming again that the hysteresis is origin from the ferroelectricity of FeFETs, more detailed explanation can be seen in Supplementary Fig. 13a-e." to "We conducted further measurements of the transfer curves across varying scan speeds, scan ranges, V_{DS} , and temperatures. The scanning window remained largely consistent, reaffirming that the hysteresis originates from the ferroelectricity of FeFETs. More detailed explanations are provided in Supplementary Fig. 13a-e.", page 11, line 6.

"This is because at higher doping concentrations, the abundance of bulk defects creates a defect field, and then surpasses the ferroelectric field." to "Due to higher doping concentrations, the substantial increase in bulk defects generates a defect field that ultimately overwhelms the ferroelectric field.", page 11, line 17, etc.

We also have renewed some figures to make them clearer in the manuscript, such as Fig. 2, Supplementary Fig. 6, Supplementary Fig. 8, Supplementary Fig. 9, Supplementary Fig. 13, and the following is Figure 2.

Fig. R8. PFM, SHG, and the ferroelectric properties of Sn-based perovskite films.

Responses to Reviewer #2

C2: This manuscript reports ferroelectric and semiconductor properties of Sn-based perovskites. By doping with imidazole-based molecule (MBI), Liu et al discovered emergence of ferroelectricity with high remanent polarization and low coercive field. The ferroelectricity is stable under bias voltage, and the origin rises from symmetry breaking by hydrogen bonding between phenylethylammonium and nitrogen of MBI. The ferroelectric transistors demonstrate low subthreshold swing of 80 mV/dec. It is our opinion that the discovery of ferroelectric behavior in Sn-based perovskites, which were applied to various high-performance field-effect transistors is very interesting, and the presented ferroelectric data are quite convincing, but the mechanism requires slightly more analysis. I therefore consider this manuscript as suitable for publication in Nature Communications, only after provided an extensive reply to the following comments and inquires.

R2: We are grateful to the reviewer for pertinent and insightful comments on our manuscript. We have addressed all comments and made necessary revisions to the manuscript based on suggestions of the reviewer. More details are described in the following point-by-point responses.

The incorporation of MBI molecules, characterized by large dipole moments, facilitates the formation of intermolecular hydrogen bonds between the nitrogen atoms of MBI and the hydrogen atoms of PEA in the 2D perovskite structure, thereby promoting the rotational motion of PEA. Concurrently, the non-coincidence of positive and negative charge centers in the 2D/3D perovskite results in broken spatial symmetry and energy level splitting. Additionally, the introduction of MBI reduces Sn vacancies, leading to a lower carrier concentration and mitigating the carrier shielding effect on the ferroelectric field, thereby enhancing ferroelectricity.

We have found DFT calculation work similar to perovskite ferroelectricity caused by the rotation of A-site cation to support our work. Wang Fei et al. demonstrated that AMP cations contribute nearly the entirety of the total polarization within (AMP)PbI₄ using DFT calculations. This is attributed to the quadrilinear coupling of order parameters, which involve two large-amplitude AMP cation rotation modes, a significant octahedral framework rotation, and ferroelectric polarization. This finding strongly suggests that the organic cations rotation could potentially facilitate emergence of ferroelectric polarization, thereby inducing an uncompensated electric polarization [*npj Computational Materials* **6**, 183 (2020)]. A theoretical study has explored the Rashba effect in 2D MAPbI₃, proposing that the distortion degree of the [PbI₆]⁻ octahedron is a key determinant of the Rashba splitting magnitude. Beichen Liu et al. found that the orientation and nature of various ligands can significantly influence the extent of Rashba splitting, with PEA ligands notably enhancing its magnitude by DFT method [*Phys. Chem. Chem. Phys.* **24**, 27827–27835 (2022)]. Machteld E. Kamminga et al. discovered that in PEA₂MnCl₄, the PEA cations undergo rotation in response to temperature changes. This rotation breaks inversion symmetry independently of any buckling in the inorganic lattice, distinguishing from mechanism where the displacement of organic molecules induces polarization. Remarkably, they observed that the resulting polar axis is oriented out-of-plane [APL Mater. **6**, 066106 (2018)].

C2.1. The authors state that it is difficult to develop a thin-film with robust ferroelectricity and excellent semiconductor properties in the first paragraph of the introduction. In the second paragraph of the introduction, the authors state that in Sn-based perovskite semiconductors, the high carrier concentration causes the internal electrical field to be not effectively screened, weakening ferroelectric polarization. The authors should give more detailed reason behind this phenomenon with supporting references, which eventually links to the success of achieving both in this manuscript. Also, the high carrier concentration of Sn-based perovskite semiconductors is often reduced by engineering of the Sn vacancy. Please comment on the efficiency of this method for arising ferroelectricity.

R2.1: Point 1: High carrier concentration can screen the ferroelectric polarization field, thereby hindering the observation of ferroelectricity. Zhou et al discovered they can't demonstrate the conventional remnant P-E hysteresis loop of α -In₂Se₃ by the Sawyer-Tower method due to charge screening [*Nano Lett.* **17**, 5508–5513 (2017)]. Kwon et al found that thicker SnS films exhibit a markedly diminished ferroelectric response since the screening effects induced by the higher concentration of charge carriers [*ACS Nano* **14**, 7628–7638 (2020)]. The reason behind this phenomenon is studied by Tomioka et al, who discovered (Sr, Ca or Ba)TiO₃ introduces carriers through Nb⁵⁺ ion doping, with the doping concentration ranging from approximately 0.05% to 0.5%. Within this range, the resistance decreases while ferroelectricity is maintained. However, when the carrier concentration exceeds a critical threshold, the increased doping completely screens the ferroelectric polarization, leading to the loss of the fundamental characteristic of ferroelectric symmetry breaking [*NPJ Quantum Materials* **7**, 111 (2022)]. The more detailed reasons and supporting references behind this phenomenon are given in the second paragraph of the introduction.

Revision: In the revised manuscript, we have added the following sentence “High carrier concentration can screen the ferroelectric polarization field, thereby hindering the observation of ferroelectricity. The conventional remnant P-E hysteresis loop of α -In₂Se₃ is difficult to demonstrate by the Sawyer-Tower method due to charge screening. Thicker SnS films exhibit a markedly diminished ferroelectric response since the screening effects are induced by the higher concentration of charge carriers. The reason behind this phenomenon is studied using (Sr, Ca or Ba)TiO₃ with doping ions, who discovered (Sr, Ca or Ba)TiO₃ introduces carriers through Nb⁵⁺ ion doping, with the doping concentration ranging from approximately 0.05% to 0.5%. Within this range, the resistance decreases while ferroelectricity is maintained. However, when the carrier concentration exceeds a critical threshold, the increased doping completely screens the ferroelectric polarization, leading to the loss of the fundamental characteristic of ferroelectric symmetry breaking.”, page 3, line 17, and reference 12, 16, 17 to the part “introduction” for linking to the success of achieving robust ferroelectricity and excellent semiconductor properties in this manuscript.

Point 2: Thanks for your careful and constructive comments to improve our work. Our findings indicate that the concentration of Sn vacancies significantly influences the emergence of ferroelectricity. Upon the introduction of MBI, the Sn²⁺ concentration increased (Fig. R9c and R9d), lowering Sn vacancies, and carrier concentration decreased (Fig. R9a), thereby facilitating the development of ferroelectricity (Fig. R9b).

Fig.R9. Relationship between regulation of carrier concentration by engineering of the Sn vacancy and arising of ferroelectricity.

Revision: In the revised manuscript, we have added the following sentence ‘**The reduction of Sn vacancy reduces the carrier concentration and is conducive to the generation of ferroelectricity.**’, page 12, line 2, to the part “High-performance transistors based on the ferroelectric perovskite semiconductors” for explaining the relationship between Sn vacancy and ferroelectricity in Sn-based perovskite film.

C2.2. The authors demonstrate an increase in crystallinity after MBI doping through increase in peak intensity of XRD. While this applies for Fig. 1, Supplementary Fig. 2 shows increased intensity for certain doping percentages of MBI: 0.3, 0.5 4.3 mol%. Is there a specific reason behind this trend? Also, PL and SHG intensities in Supplementary Fig. 7 show a comparable trend of intensities according to dopant molar percentage with each other, however, the trend seems to differ with XRD. XRD and PL intensity often show similar relationships as they are related to film quality and crystallinity. Please comment on this difference.

R2.2: Point 1: Thanks for your constructive comments on improving our work. There is another trend extracted in XRD after the introduction of MBI molecular. We extracted FWHM of each crystal plane from XRD curves in Supplementary Fig. 2. Compared with pristine, the FWHW of MBI doped perovskite film decreases on (100), denoting crystallinity has been improved. The FWHM is provided in Supplementary Table 1.

Table.R1. Fullwidth at half-maximum (FWHM) of (100) crystal plane in XRD of perovskite films with different MBI concentrations.

MBI (mol%)	FWHM	(100)
0		0.168
0.1		0.151
0.3		0.136
0.5		0.136
2.0		0.134
4.3		0.130

Revision: In the revised manuscript, we have added the following sentence ‘Compared with pristine, the FWHW of MBI doped perovskite film decreases on (100), denoting crystallinity has been improved.’, page 5, line 12, to the part “Enhanced crystallinity of the perovskite semiconductor films doped with MBI molecule” for clarifying the enhancement of crystallinity after MBI-doping. We have added Fullwidth at half-maximum (FWHM) of (100) crystal plane in Supplementary Table S1.

Point 2: After the introduction of MBI, the energy levels of the Sn-based perovskite undergo splitting, as evidenced by the emergence of a shoulder peak in the photoluminescence (PL) spectrum. The broad absorption band of the perovskite contributes to a reabsorption effect, where photons emitted from shallower energy states are reabsorbed by deeper states [*J. Phys. Chem. C* **120**, 29432–29438 (2016)]. As a result, while the overall crystallinity remains high as confirmed by XRD measurements, the PL intensity does not show a corresponding enhancement.

Revision: In the revised supporting information, we have added the following sentence ‘After the introduction of MBI, the energy levels of the Sn-based perovskite undergo splitting, as evidenced by the emergence of a shoulder peak in the photoluminescence (PL) spectrum. The broad absorption band of the perovskite contributes to a reabsorption effect, where photons emitted from shallower energy states are reabsorbed by deeper states. As a result, while the overall crystallinity remains high as confirmed by XRD measurements, the PL intensity does not show a corresponding enhancement.’, page 15, to the extension explanation of Supplementary Fig. 10.

C2.3. The AFM images of pristine and MBI doped thin-films are discussed with increase in film crystallinity. Through AFM images, the thin-film quality looks similar. The authors could comment further on this, provided with SEM images of the two thin-films.

R2.3: According to your suggestion, we provided SEM images. After introducing MBI molecules, the crystallinity of grains increased, mainly reflected in the increase of grain size and longitudinal growth.

Fig.R10. The SEM images of pristine and with MBI perovskite films.

Revision: Supplementary explanation ‘After introducing MBI molecules, the crystallinity of grains increased, mainly reflected in the increase of grain size and longitudinal growth from the SEM images (Supplementary Fig. 3)’, page 6, line 1, is added to the first part of the manuscript result, and SEM images are added to Supplementary Figure 3.

C2.4. According to the mechanism presented by the authors, the ferroelectricity arises from the breaking of symmetry with MBI doping. This causes compression of the space volume and the hydrogen bonding causing PEA molecule to be twisted and rotated. This phenomenon is well supported by XPS data, however, it seems to differ with XRD data. Lattice contortion, compression or expansion can cause peak shifts in XRD measurements. Please comment on this difference.

R2.4: Thanks for your valuable comments to improve our work. We are sorry that the statement that the space volume is compressed in the manuscript is not accurate. MBI did not incorporate into the crystal lattice, instead, hydrogen bonding induced a rotation of the PEA moieties within the two-dimensional structure of the perovskite. The primary material employed was a 2D/3D perovskite composite, with the 2D component constituting 6.7 mol% of the overall material. XRD analysis revealed the characteristic crystal plane peaks associated with the 3D perovskite structure. Notably, the XRD peak positions remained unchanged, indicating that the introduction of MBI did not alter the crystal lattice structure. The peak position is provided in Supplementary Table 2.

Table.R2. Peak position of (100) crystal plane in XRD of perovskite films with different MBI concentrations.

MBI (mol%)	Peak position	(100)
0		14.08
0.1		14.08
0.3		14.08
0.5		14.08
2.0		14.10
4.3		14.10

Revision: We have deleted the expression ‘The compression of the space volume’ and added Supplementary Table S2. In the revised manuscript, we have added the following sentence ‘MBI was not incorporated into the crystal lattice. The primary material employed was a 2D/3D perovskite composite, with the 2D component constituting 6.7 mol% of the overall material. XRD analysis revealed the characteristic crystal plane peaks associated with the 3D perovskite structure.’, page 5, line 16, to explain why the peak did not shift in XRD measurements. Additionally, we emphasize that the rotation of PEA predominantly occurs within the 2D perovskite structure, adding the words ‘within 2D perovskite’, page 9, line 2.

C2.5. While the degree of absorption can indeed vary due to the absorptive properties of the thin film, it is also a parameter that can change significantly with the thickness of the film. Therefore, when claiming an increase in absorption, it is imperative to specify the thickness of the thin film as well.

R2.5: Thanks for your insightful comments to improve our work. The thickness of pristine and with MBI film used for absorption is 38 nm, and this data is provided in Supplementary Fig. 8a.

Revision: In the revised manuscript, we have added ‘film thickness = 38 nm’ to the title of Figure 1 and the following sentence ‘The film thickness measured by the step profiler (Zeptools JS100A) is 38 nm’, page 14, line 1, to the part “Preparation of perovskite films”.

C2.6. *Could the authors provide GIWAXS data for thin films with different amounts of MBI? Since there is no clear trend in the crystallinity based on the XRD data, it is suggested to include GIWAXS data for each amount of MBI in the supplementary information to support the claims of improved crystallinity.*

R2.6: Thanks for your valuable comments to improve our work. We have provided other GIWAXS data to support the claims of improved crystallinity after the introduction of MBI molecular. While the pristine perovskite film displays broad Debye-Scherrer rings indicative of random orientation within the perovskite grains, the other perovskite films after the introduction of MBI molecular exhibit distinct Bragg points, signifying a high degree of crystal orientation. It is further proved that the introduction of MBI enhances the crystallinity of the films.

Fig.R11. The GIWAXS data of Sn-based perovskite films with different doping concentrations of MBI molecule: (a) Pristine, (b) 0.1 mol%, (c) 0.3 mol%, (d) 0.5 mol%, (e) 2.0 mol%, (f) 4.3 mol%.

Revision: In the revised manuscript, we have added the sentence ‘The other GIWAXS results with various doping concentrations are presented in Supplementary Fig. 4, suggesting improved crystallinity after the introduction of MBI molecular.’, page 6, line 9, to explain the enhanced crystallinity after doping MBI in the part of “Enhanced crystallinity of the perovskite semiconductor films doped with MBI molecule”, and the other GIWAXS data has been included in Supplementary Figure 4.

C2.7. In Supplementary Figure 5, P-V test data for the pristine sample is missing. To clearly demonstrate the effect of MBI on the induction of ferroelectric properties, please include data for the pristine samples as well.

R2.7: Thanks for your suggestion. We initially did not provide the P-V curve for the original sample because its I-V characteristics lacked a polarization reversal current, rendering the P-V measurement irrelevant. However, in line with your suggestion, and to more clearly illustrate the impact of MBI molecule introduction on ferroelectric properties, we now present the P-V curve of the initial sample for comparison. This allows for a more comprehensive understanding of the modifications brought about by MBI incorporation. Supplementary Figure 5 (revised to Supplementary Fig. 8) has been revised.

Fig.R12. P-V test results of ferroelectric analysis tester of the Sn-based perovskite films with different doping concentrations of MBI molecule.

Revision: We have added the pristine P-E loop in Supplementary Figure 8h. In the revised Supporting information, we have added the sentence ‘The pristine I-V curve (Supplementary Fig. 8b) does not exhibit a reversal peak in the polarization current, and the pristine P-V curve (Supplementary Fig. 8h) presents an annular shape, indicating that the perovskite film is not ferroelectric.’ to the description of Supplementary Figure 8.

C2.8. According to the mechanism proposed by the authors, MBI and PEA interact strongly via hydrogen bonds, causing PEA to adopt a twisted and rotated structure. Based on this, I would expect MBI to inhibit the incorporation of PEA into the perovskite lattice, potentially increasing the number of defects at grain boundaries or surfaces. This trend of increased defects can be inferred from the transfer characteristics shown in Supplementary Figure 10, which vary with the MBI content. How significant do you consider the contribution of these defects to the hysteresis observed in the transfer curves? If you claim that the contribution of charge carrier trapping at these defects is minimal, please substantiate this with time-resolved photoluminescence (TRPL) or further electrical characterization that demonstrates minimal impact from trapping. Additionally, the schematic diagram illustrating the structure shows PEA's benzene ring oriented towards the perovskite lattice, which is likely incorrect since NH^{3+} typically binds to the A cation site of the perovskite. Please correct this in the diagram.

R2.8: Point 1: Thanks for your constructive comments to improve our work. As the concentration of MBI increases, a small amount facilitates the effective electrical field, while higher concentrations promote carrier trapping. Upon the introduction of MBI molecules, ferroelectricity emerges in the perovskite material. However, this process also introduces defects. Your observation regarding defect formation at the surface or grain boundaries is particularly insightful. As the concentration of MBI molecules in the perovskite layer increases, the transfer curve of the

resulting transistor exhibits notable changes. This behavior is primarily driven by the interplay between the ferroelectric field and the defect field, which exert opposing influences on the transfer curve. We conducted time-resolved photoluminescence (TRPL) experiments, observing that as the concentration of MBI increases, the average carrier lifetime initially rises and subsequently declines, mirroring the trend in film quality.

Initially, as MBI concentration increases, the ferroelectricity of the perovskite material emergence, following the increase of average carrier lifetime, and the appearance of clockwise ferroelectric hysteresis in the transistor transfer characteristic curve coincides with an expansion of the memory window. Notably, when 0.5 mol% MBI is introduced, the ferroelectric hysteresis window in the transfer curve reaches its maximum which the ferroelectric field outweighs the defect field, highlighting the optimal balance between ferroelectricity and defect influence, meanwhile, the average carrier lifetime reaches the highest point. With the continued increase in MBI concentration, an excess of defects leads to enhanced defect trapping and release. Consequently, the transistor transfer characteristic curve exhibits anticlockwise, defect-dominated hysteresis. Simultaneously, the reduction in average carrier lifetime further indicates an intensification of defect-related effects.

Fig.R13. TRPL results of the Sn-based perovskite films with different doping concentrations of MBI molecule.

Revision: In the revised manuscript, we have added the sentence ‘whose average carrier lifetime of the perovskite layer reaches the highest point compared with other doping ratios (Supplementary Fig. 5).’, page 11, line 16, and ‘Meanwhile, TRPL showed the average carrier lifetime initially rises and subsequently declines, mirroring the trend in film quality (Supplementary Fig. 5).’, page 11, line 20, to the part of “High-performance transistors based on the ferroelectric perovskite semiconductors”.

Point 2: According to your suggestion, we have corrected the diagram.

Fig.R14. Schematic diagram of ferroelectric generation mechanism.

C2.9. In Figure 4, the x-axis of the PL data appears to represent energy in eV, but it is incorrectly labeled as wavelength. Please check and correct this notation. Additionally, Supplementary Figure 7 shows that the introduction of MBI results in the formation of a shoulder peak in the PL results due to Rashba splitting. However, the intensity of this shoulder peak does not exhibit a clear trend with increasing amounts of MBI. Could you explain why?

R2.9: Point 1: Thank you for your reminder. We have changed the label of the horizontal axis of Figure 4 of the PL data to energy.

Fig.R15. PL, CPL, and XPS spectra of the Sn-based perovskite semiconductor films.

Point 2: The emergence of the shoulder peak in the PL spectrum is primarily attributed to energy band splitting caused by the Rashba effect, rather than being a direct measure of ferroelectricity, which is more closely linked to the degree of spatial symmetry breaking. Compared to the pristine film, the perovskite exhibits pronounced band splitting in the PL spectrum following the introduction of MBI. As the MBI concentration increases, the second harmonic generation (SHG) signal of the perovskite films progressively rises, peaking at 0.5 mol% MBI, before gradually decreasing—mirroring the trend observed in ferroelectric behavior.

C2.10. Based on the results from the output curve measurements, it has been concluded that the devices show good contact behavior. Please try using techniques such as the transmission line method (TLM) to accurately evaluate the contact resistance.

R2.10: We measured the resistance of transistors with different channel lengths and extracted the contact resistance by TLM method. The extracted contact resistance is $347 \text{ } \Omega \cdot \text{cm}$, which once again proves the good contact behavior of the device.

Fig.R16. Extraction of with MBI perovskite transistor contact resistance by TLM method.

Revision: This contact resistance is supplemented in the transistor part of the manuscript ($347 \text{ } \Omega \cdot \text{cm}$), and the figure is provided in Supplementary Figure 12. In the revised manuscript, we have added the sentence ‘and the contact resistance of the transistor with MBI was measured and determined to be $347 \text{ } \Omega \cdot \text{cm}$ (Supplementary Fig. 12).’, page 11, line 7.

C2.11. The authors suggested that an excess amount of MBI would lead to a significant increase in bulk defects. Could you specify what types of bulk defects are anticipated?

R2.11: It is hypothesized that the predominant bulk defect is a PEA vacancy, which may passivate Sn vacancies [*Joule* **2**, 2732–2743 (2018), *Adv. Energy Mater.* **8**, 1702019 (2018), *Energy Technol.* **9**, 2100176 (2021)], allowing for the assessment of PEA incorporation by examining Sn vacancy behaviour. XPS results reveal that with increasing MBI incorporation, the concentration of divalent tin vacancies initially rises and then declines. Excessive MBI introduction inhibits PEA incorporation into the two-dimensional structure, leading to the formation of PEA vacancies.

Fig.R17. the X-ray photoelectron spectroscopy (XPS) (Sn 3d) of Sn-based perovskite films with different doping concentrations of MBI molecule: (a) Pristine, (b) 0.1 mol%, (c) 0.3 mol%, (d) 0.5 mol%, (e) 2.0 mol%, (f) 4.3 mol%.

Revision: In the revised manuscript, we have added the sentence ‘The predominant bulk defect is a PEA vacancy, which may passivate Sn vacancies, allowing for the assessment of PEA incorporation by examining Sn vacancy behaviour. XPS results reveal that with increasing MBI incorporation, the concentration of divalent tin vacancies initially rises and then declines (Supplementary Fig. 15). Excessive MBI introduction inhibits PEA incorporation into the two-dimensional structure, leading to the formation of PEA vacancies. The reduction of Sn vacancy reduces the carrier concentration and is conducive to the generation of ferroelectricity.’ to explain the types of defects in the part of “High-performance transistors based on the ferroelectric perovskite semiconductors”, and the XPS Sn 3d spectrum has been included in Supplementary Figure 15.

C2.12. How did the authors determine the V_{th} and the SS? While Figure 5b shows minimal changes in SS and V_{th} , the manuscript indicates a substantial difference. If extrapolation was used, please show how it was derived from the graph. If another method was employed, please also describe that method.

R2.12: Thanks for your suggestion, we have provided the V_{th} and SS extraction methods for both pristine and MBI-doped perovskite transistors, which the method derives from [Nat. Commun. 10, 3037 (2019), Nat. Electron. 5, 416–423 (2022)]. The threshold voltage is determined from the transfer characteristic curve in linear coordinates at the point where the current extension line intersects the X-axis. The subthreshold swing is extracted using the following formula: $SS = dV_G / d(\lg I_D)$, where V_G means gate voltage, I_D means channel current.

Fig.R18. V_{th} and SS extraction methods for both pristine and MBI-doped perovskite transistors.

Revision: In the revised manuscript, we have added the sentence ‘which extraction method was provided in Supplementary Figure 16’, page 12, line 10, and Supplementary Figure 16.

C2.13. The transfer curves measured at low temperatures display only hysteresis in the subthreshold region, while virtually no hysteresis is observed in other regions. This makes it challenging to assert that there is no change in the hysteresis window with temperature. Please check and correct this. Additionally, why do you think the hysteresis observed in the transfer curves at low temperatures is significantly smaller than that at room temperature? Furthermore, according to Supplementary Figure 9f, the transfer curve of a transistor containing 0.5 mol% measured near room temperature at 290K shows considerable differences in hysteresis compared to what is displayed in Figure 5b. What is the reason?

R2.13: Point 1: Thanks for your valuable comments to improve our work.

Revision: We reviewed this section and revised the phrase "no change in the hysteresis window with temperature" to "no significant change in the hysteresis window at varying temperatures within the subthreshold region".

Point 2: The hysteresis observed in the transfer curve of ferroelectric perovskite transistors primarily arises from defect trapping and ferroelectric polarization. The extent of defect trapping is temperature-dependent [*ACS Nano* **10**, 9543–9549 (2016)]; at lower temperatures, the thermal activation energy is insufficient for carriers to escape from defect sites after being captured and certain defect states may become "frozen", meaning that their ability to interact with carriers (i.e., to capture or release them) is diminished [*Solid-State Electronics* **49**, 545–553 (2005)]. This is often observed in semiconductors where deep-level defects, which require significant energy for interaction, become inactive at low temperatures [*Adv. Sci.* **4**, 1700183 (2017)]. Consequently, the impact of defect capture and release diminishes as temperature decreases. In contrast, the effect of ferroelectric polarization remains largely independent of temperature [*Adv. Mater.* **32**, 2004813 (2020)]. Therefore, at low temperatures, the influence of the defect field is reduced compared to that at room temperature, allowing the ferroelectric polarization effect to become more prominent. This part has been added to the description of Supplementary Figure 13.

Revision: In the revised Supporting information, we have added the sentence ‘The hysteresis observed in the transfer curve of ferroelectric perovskite transistors primarily arises from defect trapping and ferroelectric polarization. The extent of defect trapping is temperature-dependent; at lower temperatures, the thermal activation energy is insufficient for carriers to escape from defect sites after being captured and certain defect states may become "frozen", meaning that their ability to interact with carriers (i.e., to capture or release them) is diminished. This is often observed in semiconductors where deep-level defects, which require significant energy for interaction, become inactive at low temperatures. Consequently, the impact of defect capture and release diminishes as temperature decreases. In contrast, the effect of ferroelectric polarization remains largely independent of temperature. Therefore, at low temperatures, the influence of the defect field is reduced compared to that at room temperature, allowing the ferroelectric polarization effect to become more prominent.’ to the description of Supplementary Figure 13.

Point 3: This phenomenon can be attributed to variations in the voltage scanning range. The minimum voltage presented in Fig. 5b is 4V, whereas in Supplementary Fig. 9f (revised to Supplementary Fig. 13f), it decreases to 0V. Hysteresis predominantly manifests in the subthreshold region of the transfer curve. However, as the scanning range expands and reaches the saturation region, the degree of hysteresis decreases. With the voltage nearing 0 V, the increasing carrier concentration in the perovskite layer begins to shield the ferroelectric field. The saturation region with high carrier concentration is more sensitive to charge capture and release processes, while the subthreshold region has a smaller carrier concentration, resulting in a greater impact of defect capture and release on the transfer curve in the saturation region.

Revision: In the revised Supporting information, we have added the sentence ‘The minimum voltage presented in Fig. 5b is 4V, whereas in Supplementary Fig. 13f, it decreases to 0V. Hysteresis predominantly manifests in the subthreshold region of the transfer curve. However, as the scanning range expands and reaches the saturation region, the degree of hysteresis decreases. With the voltage nearing 0 V, the increasing carrier concentration in the perovskite layer begins to shield the ferroelectric field. The saturation region with high carrier concentration is more sensitive to charge capture and release processes, while the subthreshold region has a smaller carrier concentration, resulting in a greater impact of defect capture and release on the transfer curve in the saturation region.’ to the description of Supplementary Figure 13.

Responses to Reviewer #3

C3: I co-reviewed this manuscript with one of the reviewers who provided the listed reports. This is part of the Nature Communications initiative to facilitate training in peer review and to provide appropriate recognition for Early Career Researchers who co-review manuscripts.

R3: We are grateful to the reviewer for pertinent and insightful comments on our manuscript. We have addressed all comments and made necessary revisions to the manuscript based on suggestions of the reviewer.

Responses to Reviewer #4

C4: Ferroelectric semiconductors offer the dual advantages of switchable polarization and semiconductor transport characteristics, making them highly promising for applications in ferroelectric transistors and nonvolatile memory devices. However, preparing Sn-based perovskite films with robust ferroelectric and semiconductor properties remains challenging. In this study, authors have shown that doping Sn-based perovskite films with 2-methylbenzimidazole (MBI) effectively transforms them into ferroelectric semiconductor films through molecular reconfiguration. These reconfigured films exhibit a high remanent polarization (P_r) of 23.2 $\mu\text{C}/\text{cm}^2$ and a low coercive field (E_c) of 7.2 kV/cm. The emergence of ferroelectricity is attributed to the enhancement of hydrogen bonds following imidazole molecular doping, which breaks spatial symmetry and causes the positive and negative charge centers to become non-coincident. Notably, transistors based on these perovskite ferroelectric semiconductors demonstrate a low subthreshold swing of 80 mV/dec, further highlighting the advantages of introducing ferroelectricity.

This work presents a novel approach to realizing Sn-based ferroelectric semiconductor films for electronic device applications. The manuscript is well-written, short, and concise, with no ambiguity. I recommend it for publication in Nature Communications.

R4: We are very grateful for your affirmation of our work. We have addressed all comments and made necessary revisions to the manuscript based on suggestions. More details are described in the following point-by-point responses.

C4.1: The origin of ferroelectricity is not discussed properly. I suggest the authors revise the discussion of ferroelectric origin with further DFT calculations and/or by citing previous DFT-based published papers.

R4.1: Thanks for your insightful and constructive comments to improve our work. We have found DFT calculation work similar to perovskite ferroelectricity caused by the rotation of A-site cation to support our work.

Wang Fei et al. demonstrated that AMP cations contribute nearly the entirety of the total polarization within (AMP)PbI₄ using DFT calculations. This is attributed to the quadrilinear coupling of order parameters, which involve two large-amplitude AMP cation rotation modes, a significant octahedral framework rotation, and ferroelectric polarization. This finding strongly suggests that the organic cations rotation could potentially facilitate the emergence of ferroelectric polarization, thereby inducing an uncompensated electric polarization [*npj Computational Materials* **6**, 183 (2020)]. A theoretical study has explored the Rashba effect in 2D MAPbI₃, proposing that the distortion degree of the [PbI₆]⁻ octahedron is a key determinant of the Rashba splitting magnitude. Beichen Liu et al. found that the orientation and nature of various ligands can significantly influence the extent of Rashba splitting, with PEA ligands notably enhancing its magnitude by the DFT method [*Phys. Chem. Chem. Phys.* **24**, 27827–27835 (2022)]. Machteld E. Kamminga et al. discovered that in PEA₂MnCl₄, the PEA cations undergo rotation in response to temperature changes. This rotation breaks inversion symmetry independently of any buckling in the inorganic lattice, distinguishing it from the mechanism where the displacement of organic molecules induces polarization. Remarkably, they observed that the resulting polar axis is oriented out-of-plane [APL Mater. **6**, 066106 (2018)].

Revision: In the revised manuscript, we have added the sentence ‘This method of generating ferroelectricity is validated through density functional theory (DFT) calculations, which offer a robust certification for predicting and confirming ferroelectricity’, page 9, line 15, into the part of

“Origin of ferroelectricity in Sn-based perovskite semiconductors doped with MBI molecule” and the references **26, 27, 28** to support the mechanism we mentioned.